# High-fidelity long-read sequencing of an avian herpesvirus reveals extensive intrapopulation diversity in tandem repeat regions

Alejandro Ortigas-Vasquez[1], Christopher D. Bowen[1], Daniel W. Renner[1],
Susan J. Baigent[2], Yaoyao Zhang[2], Yongxiu Yao[2], Venugopal Nair[2], David A. Kennedy[1],
Moriah L. Szpara[1,3]*

1 Departments of Biology, 2 Viral Oncogenesis Group, The Pirbright Institute, Woking, United Kingdom,
3 Biochemistry and Molecular Biology, Center for Infectious Disease Dynamics and the Huck Institutes of
the Life Sciences, Pennsylvania State University, University Park, Pennsylvania, United States of America

* moriah@psu.edu

## Abstract

Comparative genomic studies of Marek's disease virus (MDV) have suggested that attenuated and virulent strains share >98% sequence identity. However, these estimates fail to account for variation in regions of the MDV genome harboring tandem repeats. To resolve these loci and enable assessments of intrapopulation diversity, we used a PacBio Sequel II platform to sequence MDV strains CVI988/Rispens (attenuated), HPRS-B14 (virulent), Md5 (very virulent) and 675A (very virulent plus). This approach enabled us to identify patterns of variation in tandem repeat regions that may contribute to the known phenotypic differences between these strains, including the proline-rich regions of the *meq* oncogene (Meq-PRR) and MDV049/UL36 (UL36-PRR), the multiple telomeric repeats (mTMR) region of the *a*-like sequence, the promoter region of the latency-associated transcript (LAT), and the MDV006.5/MDV075.2 transcripts. We also found CVI988/Rispens variants showing a 4.3-kb deletion in the Unique Short (US) region, resulting in the loss of SORF1, SORF2, US1, US10, SORF3, and US2. Despite the conventional wisdom that MDV harbors little genomic diversity even when compared to other herpesviruses, we found MDV tandem repeat regions to be highly variable both within individual samples and across strains. In addition to providing a foundation for future studies seeking to explore a potential link between MDV tandem repeats and phenotypic traits like virulence and attenuation, these findings provide detailed support for the premise that DNA viruses can harbor high levels of within-sample diversity in tandem repeat regions.

## Author's Summary

Marek's disease virus (MDV) is a globally important avian pathogen, with annual losses of over $1 billion USD for the poultry industry. In prior decades, circulating MDV strains have evolved to overcome the protection of commercial

**Data availability statement:** All relevant data are within the manuscript and its Supporting information files. Viral genomes have been deposited to GenBank, as Accessions PV035744, PV035745, PV035746, PV035747. Additional data related to this manuscript (e.g. alignment files) have been deposited at the public repository Scholarsphere: doi:10.26207/e0fh-0c45

**Funding:** This work was supported by NSF-NIH EEID award 1 R01 GM140459 (DK, MS), and the Biotechnology and Biological Sciences Research Council (BBSRC) grants BBS/E/I/00007038, BBS/E/PI/23NB0003, and BB/V017748/1 (VN, YY). The funders had no role in study design, data collection and analysis, decision to publish, or preparation of the manuscript.

**Competing interests:** The authors have declared that no competing interests exist.

vaccines. The current gold standard vaccine, known as CVI988 or Rispens, has successfully protected commercial flocks since the 1970s. However, there are reports of new MDV strains that may be capable of overcoming this vaccine. To better understand the genetic differences that distinguish attenuated vaccine strains from the virulent circulating strains capable of causing disease, we used new sequencing methods that rely on longer DNA fragments to reconstruct and to examine previously inaccessible viral genome regions that contain large stretches of repetitive DNA. Using this approach, we identified new patterns of genetic variation in these repeats, several of which are found in genomic regions previously associated with MDV virulence. These data lead to new testable hypotheses about the potential association of patterns detected in these repeat regions with virulence, and highlight new loci that could be used to distinguish vaccine strains from circulating disease-causing strains. The application of these methods to other herpesviruses may reveal similar variation in repetitive regions, which have been "hidden" by older sequencing methods that rely on shorter DNA fragments.

## Introduction

Marek's disease virus (MDV) is an alphaherpesvirus that causes a lymphoproliferative and demyelinating disease in poultry (*Mardivirus gallidalpha 2*, Genus *Mardivirus*; Family *Herpesviridae*) [1]. Since the first description of Marek's disease (MD) in 1907, MDV has undergone three major shifts in virulence [2–4]. As a result, currently circulating strains of MDV have been classified into four main groups, or "pathotypes", depending on their ability to bypass the protection induced by commercial vaccines: mild (m), virulent (v), very virulent (vv) or very virulent plus (vv+) [5]. The current "gold standard" commercial vaccine against MDV is a live-attenuated *Mardivirus gallidalpha 2* vaccine known as CVI988 or Rispens. This vaccine was produced through serial passage in duck-embryo fibroblast cells, and has successfully protected commercial flocks for the past three decades [6,7]. However, a number of studies have recently reported the emergence of MDV strains capable of overcoming the protection conferred by the CVI988/Rispens vaccine [8–10]. To facilitate the development of novel vaccines that can protect commercial flocks against future outbreaks, there is a pressing need to understand the molecular basis of MDV attenuation and the factors contributing to the overall virulence of MDV strains [11–13]. In addition, a better understanding of the genetic and genomic diversity of MDV vaccines and circulating strains could help to inform the development of vaccine candidates for human alphaherpesviruses such as herpes simplex virus (HSV) and varicella-zoster virus (VZV).

The MDV genome is 170–181-kb in length and shows a structure typical of alphaherpesviruses, which consists of a Unique Long (UL) and a Unique Short (US) region (~115-kb and ~12-kb in length, respectively) that are each flanked by inverted structural repeat regions. Comparative analyses of MDV consensus genomes performed in the last two decades have suggested that the CVI988/Rispens vaccine is

antigenically and genetically >98% identical to virulent strains [14]. However, these estimates fail to account for variation in 5 MDV genomic features associated with long stretches of tandem repeats, including: the MDV006.5/MDV075.2 transcripts, the proline-rich region of MDV049/UL36 (UL36-PRR), the multiple telomeric repeats (mTMR) region of the *a*-like sequence, the promoter region of the latency-associated transcript (LAT), and the proline-rich region of the *meq* oncogene (Meq-PRR) [15]. Most of these features occur in the structural repeat regions flanking the UL and US regions, giving rise to 9 distinct genomic locations that harbor tandem repeats (**Fig 1D**). While available sequencing data suggests that MDV tandem repeats can be highly polymorphic, they have been excluded from most comparative genomic studies due to the difficulty of resolving them using Illumina-based approaches [16–19]. The few studies that have attempted to resolve MDV tandem repeats relied mostly on a combination of PCR and Sanger sequencing, but such an approach is low-throughput, cost ineffective, and cannot reliably resolve repeats >800-bp in length [15,20,21]. As a result, these regions have remained relatively understudied compared to the rest of the MDV genome, and it is currently unknown whether and/or how they correlate with attenuation or virulence.

In recent years, the development of sequencing technologies capable of generating longer reads than those produced by Illumina platforms have made it increasingly practical for comparative genomic studies to account for variation in tandem repeats [22,23]. Sequencing platforms relying on Single-Molecule Real-Time (SMRT) and Nanopore technologies have already been successfully used to resolve tandem repeats >800-bp in length [24,25]. However, early iterations of long-read sequencing technologies suffered from relatively high error rates (>25%), making them unsuitable for the detection of viral genomic variants present at low frequencies (i.e., minor variants) [26]. More recently, Pacific Biosciences (PacBio) developed a high-fidelity (HiFi) long-read sequencing method known as circular consensus sequencing (CCS), which can generate reads 5–25 kb in length with error-rates comparable to Illumina platforms [27]. This improvement in long-read sequencing technology makes it feasible to thoroughly and deeply analyze herpesvirus population diversity throughout the genome, including in complex repeat regions.

In this study, we sought to take advantage of the length and accuracy of PacBio HiFi reads to resolve tandem repeat regions in the MDV genome and perform assessments of intrapopulation diversity at these loci. We used a PacBio Sequel II platform to sequence cultured viral stocks of MDV strains CVI988/Rispens (attenuated), HPRS-B14 (v), Md5 (vv) and 675A (vv+) [13]. PacBio HiFi reads were then mapped to a strain-matched, *de novo*-assembled Illumina reference genome and visualized in a genome browser to facilitate manual curation and variant calling. Using this approach, we identified several patterns of tandem repeat variation that may contribute to the known phenotypic differences between these strains [13]. The use of PacBio HiFi reads also enabled us to identify three distinct repeating units in the Meq-PRR, which differs from past descriptions of this locus. These three repeating units can account for structural differences across all Meq "isoforms", including a new variant of Meq identified as part of this study. We also detected CVI988/Rispens genomes containing a 4.3-kb deletion in the US region, which resulted in the loss of SORF1, SORF2, US1, US10, SORF3, and US2. These findings showcase the ability of PacBio HiFi reads to accurately resolve tandem repeats in herpesvirus genomes, and provide a foundation for future studies seeking to explore the potential role of MDV tandem repeat loci in virulence and attenuation.

## Methods

### Cells, virus master stocks, and virus working stocks

Primary chick embryo fibroblasts (CEF) were prepared from 10-day old embryos and maintained in M199 medium (Thermo Fisher Scientific, Waltham, MA, USA) supplemented with 5% fetal bovine serum (FBS, Sigma-Aldrich, Darmstadt, Germany), 100 units/mL of penicillin and streptomycin (Thermo Fisher Scientific), and 10% tryptose phosphate broth (Sigma).

Virus master stocks for MDV strains HPRS-B14, 675A, and Md5 (all 7th duck embryo fibroblast passage stocks) were kindly provided by Dr. A. M. Fadly (Avian Disease and Oncology Laboratory, USA). Master stocks of these viruses were

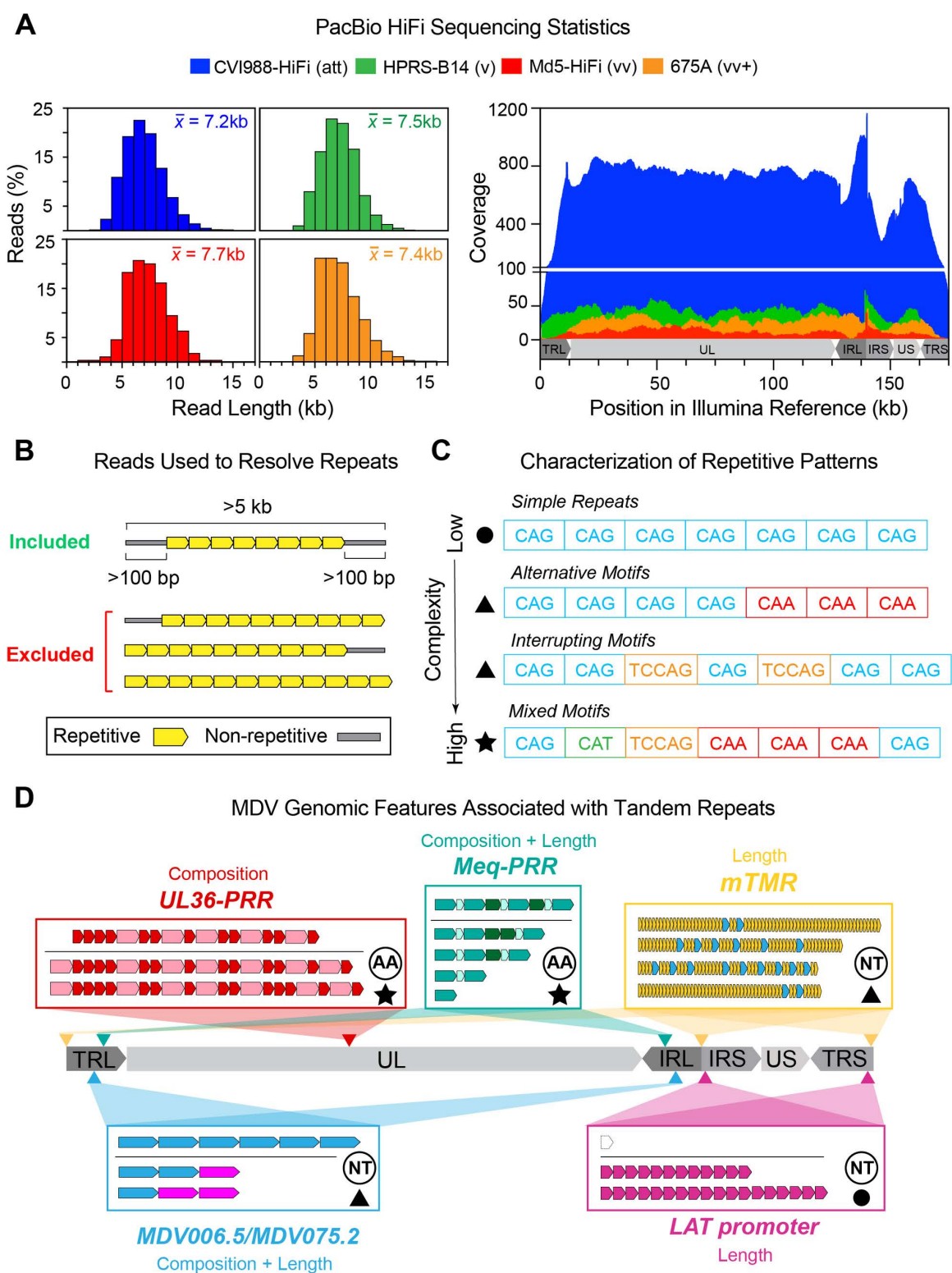

**Fig 1. Using PacBio HiFi reads to resolve tandem repeats in four MDV strains and assess their intrapopulation diversity. A)** Four MDV strains with known phenotypic differences (att = attenuated, v = virulent, vv = very virulent, vv+ = very virulent plus) were sequenced using a PacBio Sequel II platform. Histograms (left) show the cumulative length distribution of quality-filtered, MDV-specific PacBio HiFi reads used to generate viral consensus

genomes for each strain. The mean read length is displayed at the top right corner of each histogram. Coverage graph (right) shows the number of PacBio HiFi reads overlapping each position of the strain-matched *de novo*-assembled Illumina reference genome for each strain. **B)** To resolve tandem repeats, only reads >5-kb in length that fully contained the repeated motifs and extended at least 100-bp into non-repetitive sequences at either side were considered. Reads ending or starting with repeated motifs, as well as reads that only contained repetitive sequence, were excluded regardless of length. **C)** Tandem repeats can be classified based on the complexity of their repetitive patterns. Simple repeats (top, circle symbol) consist of perfect reiterations of a single repeating unit (e.g., CAG). Compound repeats (middle, triangle symbols) involve an alternative version of the repeating unit or interruptions in the repeat sequence by another motif. Complex repeats (bottom, star symbol) involve multiple alternative repeat versions and/or the presence of multiple repeating units in a variety of different patterns. **D)** The MDV genome consists of two large unique segments (Unique Long = UL, Unique Short = US), each flanked by inverted structural repeats (Terminal/Internal Repeat Long = TRL/IRL, or Short = IRS/TRS). A total of 5 genomic features are associated with tandem repeats, including: the MDV006.5/MDV075.2 transcripts, the proline-rich region of MDV049/UL36 (UL36-PRR), the multiple telomeric repeats (mTMR) region of the *a*-like sequence, the promoter region of the latency-associated transcript (LAT), and the proline-rich region of the *meq* oncogene (Meq-PRR). These features give rise to 9 distinct genomic locations in the MDV genome, indicated with arrow heads. For each genomic feature, a graphic representation of its characteristic repetitive patterns is shown (additional details in Figs 2–6). In each repetitive locus, sequence differences across MDV strains were primarily associated with length (i.e., number of repeat copies), composition (i.e., presence/absence of specific alternative motifs) or both. Where notable differences exist between the repeat patterns detected in CVI988/Rispens vs. the disease-causing strains (all repeats except mTMR), a horizontal line separates the patterns observed in the vaccine strain (above the line) from other strains (below the line). Symbol shapes indicate the complexity of the observed patterns (star = complex, triangle = compound, circle = simple). Circled abbreviations on each repeat region indicate whether the repeat was analyzed at the nucleotide (NT) or the amino acid (AA) level.

prepared as previously described [13], and then further passaged in CEF every 5–6 days to produce working stocks of virus. A commercial CVI988/Rispens vaccine was sourced in the UK (Poulvac Marek CVI vaccine; Zoetis).

### Virus cultures and DNA isolation for Illumina sequencing of MDV strains HPRS-B14 and 675A

The MDV working stocks we refer to as "HPRS-B14-Illumina" and "675A-Illumina" were derived from the master stocks as described above. DNA was prepared from approximately $5\times10^6$ cells of these working stocks (5th passage CEF stock for HPRS-B14, and 4th passage CEF stock for 675A), using the DNeasy-96 kit (Qiagen, Hilden, Germany), according to the manufacturer's instructions, and eluted in DNase-free water.

### Virus cultures and DNA isolation for PacBio HiFi sequencing

The MDV working stock we refer to as "CVI988-HiFi" was obtained from the commercial CVI988/Rispens vaccine following 2 passages in CEF with an infection level of 0.01 pfu/cell.

The MDV working stocks for HPRS-B14 and 675A, as well as the MDV stock we refer to as "Md5-HiFi", were derived from the master stocks as described above. DNA was prepared from approximately $5\times10^6$ cells of these working stocks (2nd passage CEF stock for CVI988/Rispens, 9th passage CEF stock of Md5, 5th passage CEF stock for HPRS-B14, and 4th passage CEF stock for 675A), using the DNeasy-96 kit as described above.

### DNA library preparation and Illumina sequencing

DNA libraries for MDV strains HPRS-B14 and 675A were prepared for next generation sequencing using an Illumina MiSeq platform as previously described [28–30]. Briefly, extracted DNA from MDV cultures was quantified for total DNA by Qubit (Thermo Fisher) and total viral DNA by a qPCR assay targeting viral gene pp38 [31]. Total DNA was then acoustically sheared using a Covaris M220, with settings at 60s duration, peak power 50, 10% duty cycle, at 4 °C. Sheared DNA fragments were processed for library preparation using the KAPA HyperPrep Library Amplification kit (KAPA Biosystems) according to the manufacturer's specifications. Custom MDV-specific oligonucleotide primers (Arbor Biosciences) were used to enrich for virus-specific material [32]. Enriched libraries were amplified by PCR (14 cycles) using the KAPA HiFi HotStart Library Amplification Kit (KAPA Biosystems). Finally, libraries were quantified again by Qubit and qPCR as well as a KAPA-specific qPCR (KAPA Biosystems) and TapeStation (Agilent Technologies). Libraries were then sequenced using 300 x 300 bp paired end reads v3 chemistry on an Illumina MiSeq (Illumina).

### DNA library preparation and PacBio HiFi sequencing

The SMRTbell Prep Kit 3.0 was used to prepare indexed templates according to the manufacturer's protocol (Pacific Biosciences of California, Inc.). DNA concentration was measured by Qubit and MDV genome copy number was quantified by qPCR, as above. Templates were pooled and subsequently bound to polymerase using the Sequel II Binding Kit 2.0 (Pacific Biosciences of California, Inc.). Templates were sequenced with a 30-hour movie time on a Sequel IIe using a Sequel II Sequencing Plate 2.0 (Pacific Biosciences of California, Inc.).

### Processing of illumina reads and genome assembly

MDV-specific Illumina reads were identified using Kraken2 with default settings and extracted into a separate file using a custom Python script [33]. The extracted reads were then subjected to the quality control and preprocessing step (Step 1) of our published viral genome assembly (VirGA) workflow, which performs adaptor trimming and trimming of low-quality bases [34]. These properly paired reads were then used for *de novo* assembly using metaSPAdes v3.14.0 [35]. The resulting file containing the metaSPAdes scaffolds in FASTA format served as input for the remaining steps of VirGA (Steps 3–4), which include genome linearization, annotation and post-assembly quality assessments.

### Processing of PacBio HiFi reads and genome assembly

MDV-specific PacBio HiFi reads were identified using Kraken2 with default settings and extracted into a separate file using a custom Python script. For each strain, PacBio HiFi reads were mapped to a strain-matched, full-length Illumina reference using Minimap2 v2.28 [36]. Reads with mapping quality of zero (MAPQ = 0) and supplementary alignments were removed using Samtools v1.16. Mapped reads were visualized without clipping using the Integrative Genomic Viewer (IGV) v2.12.3 [37]. In non-repetitive regions, variant calling was conducted by identifying positions with at least 2x coverage and >50% disagreement between mapped reads and the reference sequence. Illumina templates of each strain were then manually modified using Geneious Prime v2024.0.5 to generate PacBio HiFi consensus genomes [38]. To resolve repetitive loci, reads of length >5-kb that fully contained the corresponding tandem repeat region and also extended partially (>100-bp) into non-repetitive regions upstream and downstream were aligned using MAFFT v7.505 and imported into Geneious Prime for manual re-alignment (Table A in S1 File) [39]. The "Generate Consensus Sequence" tool in Geneious Prime was then used to obtain the consensus sequence for each repetitive locus. To account for low read coverage in TRL and TRS, the reverse complements of the PacBio HiFi-resolved IRL and IRS regions were used. The final full-length genomes were deposited into GenBank under accession numbers: PV035744, PV035745, PV035746 and PV035747.

### Whole-genome pairwise identity and phylogenetic analyses

Full-length PacBio HiFi consensus genomes of all four strains (n = 4) and previously published MDV consensus genomes with PubMed Identifiers (n = 39, see Table B in S1 File for list of GenBank accessions) were aligned using MAFFT. The genome alignment file is deposited at https://doi.org/10.26207/e0fh-0c45. Gaps were masked using the "Mask Alignment" tool in Geneious Prime. A maximum-likelihood (ML) tree based on the masked alignment was constructed using IQ-TREE v2.4.0 [40]. The Bayesian Information Criterion (BIC) was used to determine the most suitable substitution model. Bootstrap values were computed using 1000 replicates.

### MDV049/UL36 proline-rich region pairwise identity and phylogenetic analyses

The amino-acid sequence of the MDV049/UL36 proline-rich region (UL36-PRR) for the four strains sequenced using PacBio HiFi reads (n = 4) and for MDV strains with previously published consensus genomes with PubMed Identifiers (n = 39, Table B in S1 File) was extracted and aligned using the "Geneious Alignment" tool in Geneious Prime. The resulting

alignment was visually inspected and manually re-aligned. Strains exhibiting stretches of Ns in the UL36-PRR or completely lacking this region were excluded (n = 3). The remaining sequences (n = 41) were used to generate a ML tree using IQ-TREE. The Bayesian Information Criterion (BIC) in IQ-TREE was used to determine the most suitable substitution model. Bootstrap values were computed using 1000 replicates.

## Results

### Whole-genome sequencing of four MDV strains using high-fidelity long reads

After quality control and filtering steps, the total number of MDV-specific PacBio HiFi reads for all four strains ranged from 245 to 17,548 (Table 1). Read lengths ranged from 0.1-14 kb and averaged ~7 kb in each of the four sequenced samples (Fig 1A, left). To generate viral consensus genomes, PacBio HiFi reads were mapped to a strain-matched Illumina reference genome (Fig 1A, right). For MDV strains CVI988/Rispens and Md5, a previously published consensus genome of the same strain was used as the reference ("CVI988-UK" and "Md5-UK", respectively, see Table B in S1 File for GenBank accessions). The lack of published consensus genomes of MDV strains HPRS-B14 and 675A was addressed by generating new Illumina consensus genomes to serve as reference for each of these two strains (see Methods). The resulting alignment was then visually inspected to identify disagreements between the mapped PacBio HiFi reads and the Illumina reference sequence. As part of this process, we noticed that a high proportion (>50%) of reads mapping to tandem repeat loci contained large insertions and/or deletions (indels) relative to the reference. For tandem repeats involving a single repeating unit, indel lengths were found to correlate with discrete changes in repeat copy numbers (i.e., repeat expansions/contractions). Most reads contained indels of lengths that were exact multiples of the repeat period size. A minority of reads contained indels that were 1–3 bp away from a length that was exactly divisible by the period size. These small deviations were found to exclusively result from changes in homopolymer lengths, and were highly associated with misaligned reads. For tandem repeats involving multiple repeating units (i.e., complex repeats), motif compositions could not be inferred from indel sizes alone. To account for this discrepancy, reads of length >5-kb that fully contained the corresponding repetitive region and also extended partially (>100-bp) into non-repetitive regions upstream and downstream were extracted and manually re-aligned (Fig 1B, Table A in S1 File). For each strain, final consensus genomes were generated by manually modifying each initial reference genome to reflect and encompass all sites where >50% of the mapped PacBio HiFi reads differed from the reference. Phylogenomic analyses based on a multiple-sequence alignment with gaps excluded of the 4 newly assembled genomes alongside 39 previously published consensus genomes of MDV revealed that strain 675A is closest genetically to 648A, another vv+ strain isolated in North America. The nearest neighbor to strain HPRS-B14 (v), although still on a distinct branch, was strain CU-2 (mild), a low virulence strain (Fig A in S1 File). The new PacBio HiFi genomes of strains CVI988/Rispens (CVI988-HiFi) and Md5 (Md5-HiFi) clustered near prior sequences of the same strains. To account for tandem repeat variation in the four MDV strains sequenced using PacBio HiFi, we first characterized the observed repetitive patterns (i.e., number of repeating units involved, presence of alternative and/or interrupting motifs, etc.) (Fig 1C). We then investigated the underlying patterns of intrapopulation diversity found at each MDV tandem repeat locus in these strains (Fig 1D).

**Table 1. Sequencing metrics for MDV strains sequenced using PacBio HiFi.**

| MDV Strain | DNA Concentration (ng/$\mu$l) | # Total Reads | # MDV-specific reads | # Post-QC Reads used for assembly | Average Coverage | GenBank Accession | Consensus Genome Length (bp) |
|---|---|---|---|---|---|---|---|
| CVI988-HiFi | 44.5 | 700,124 | 18,179 | 17,548 | 606x | PV035744 | 181,283 |
| HPRS-B14 | 31.1 | 448,692 | 992 | 931 | 37x | PV035747 | 180,021 |
| Md5-HiFi | 12.2 | 587,953 | 285 | 245 | 10x | PV035745 | 181,672 |
| 675A | 4.5 | 767,371 | 638 | 569 | 24x | PV035746 | 178,979 |

## MDV strains CVI988-HiFi and 675A exhibit intrapopulation diversity in the 132-bp repeats overlapping the MDV006.5/MDV075.2 transcripts

The MDV006.5/MDV075.2 transcripts are located in Terminal Repeat Long (TRL) and Internal Repeat Long (IRL), respectively. While their function is still unknown, these transcripts have been shown to be variable in length due to partially overlapping a set of 132-bp tandem repeats [41,42]. To date, two versions of the 132-bp repeating unit have been identified [15]. The most common version contains a cytosine (based on MDV075.2) in position 67, and is typically present regardless of pathotype. However, some very virulent (vv) and very virulent plus (vv+) strains have been shown to contain an alternative version with a thymine at this position. Both nucleotides encode alanine (i.e., synonymous) in the resulting protein (**Fig 2A**). We refer to these repeated motifs as 132A (cytosine) and 132B (thymine). At the consensus level, the attenuated vaccine strain CVI988-HiFi was found to contain 10 copies of the 132A repeat (**Fig 2B**). Strains HPRS-B14 (v)

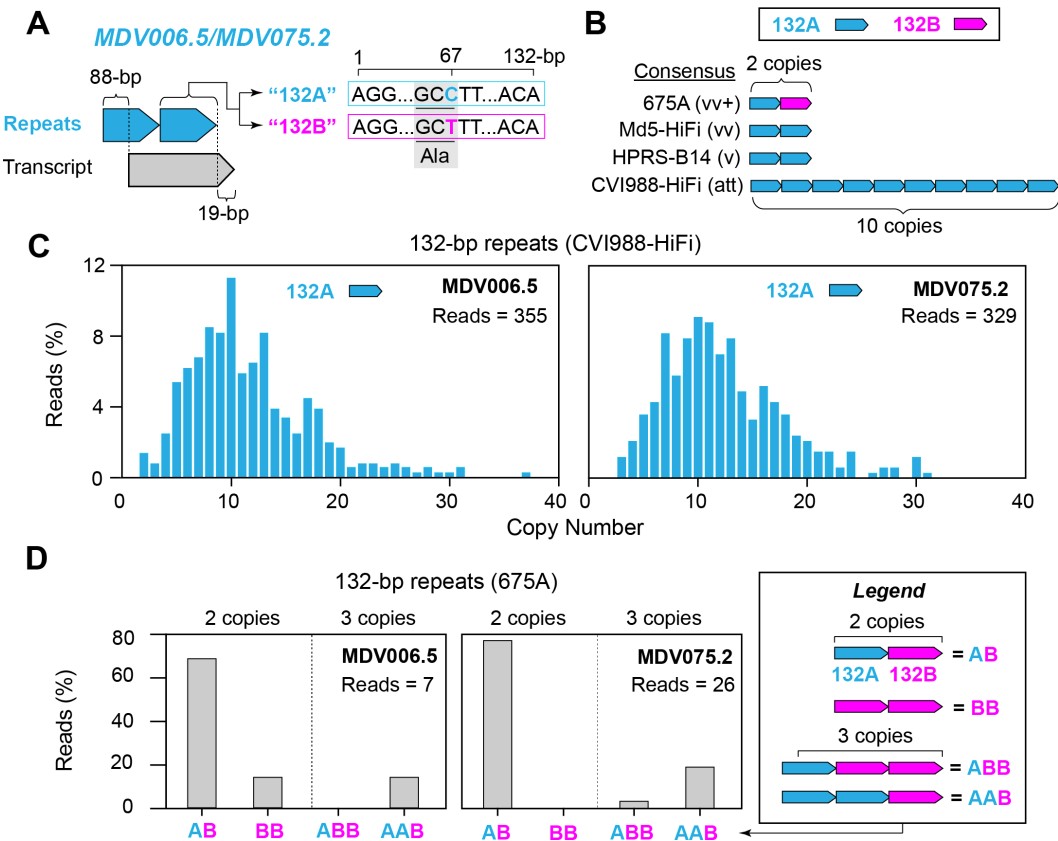

**Fig 2. High copy number heterogeneity and presence of an alternative motif in the MDV006.5/MDV075.2 transcripts are associated with phenotypic extremes. A)** The MDV006.5/MDV075.2 transcripts are found in TRL and IRL, respectively, and have been shown to vary in length due to their partial overlap with a set of 132-bp tandem repeats (light blue). Two versions of the 132-bp repeating unit are known to exist, distinguished only by a C>T mutation at nucleotide position 67 within the repeating unit (132A = light blue, 132B = fuchsia). The synonymous codons associated with this position both encode an alanine (Ala). **B)** The modal copy number was used to determine the consensus sequence of the MDV006.5/MDV075.2 transcripts for CVI988-HiFi, HPRS-B14, Md5-HiFi and 675A. **C)** PacBio HiFi reads of MDV strain CVI988-HiFi mapping to MDV006.5 and MDV075.2 exclusively contained the 132A version (light blue) of the repeating unit. Copy numbers ranged from 2-37 copies and 3-32 copies, respectively. For both loci, a modal copy number of 10 copies was supported by less than 12% of the total reads mapping to each location. **D)** For MDV strain 675A, PacBio HiFi reads mapping to these loci showed patterns of variation involving both versions of the 132-bp repeating unit, but always including the 132B version. A total of four genotypes with repeat copy numbers ranging from 2-3 were identified. These include "AB", "BB", "ABB", and "AAB", where A indicates the 132A version of the repeat (light blue) and B indicates the 132B version (fuchsia). In both MDV006.5 and MDV075.2, the modal genotype was found to be "AB".

and Md5 (vv) each contained 2 copies of the 132A repeat, while strain 675A (vv+) contained one 132A repeat followed by a 132B repeat. For CVI988-HiFi, a total of 684 PacBio HiFi reads were found to completely contain the 132-bp repeats and extend into the Unique Long (UL) region, enabling comparisons between MDV006.5 (n = 355) and MDV075.2 (n = 329) 132-bp repeat copy numbers. A total of 352 reads (99.2%) mapping to MDV006.5 and 325 reads mapping to MDV075.2 (98.8%) showed copy number variation relative to the Illumina reference. CVI988-HiFi reads mapping to MDV006.5 contained 2–37 copies of the 132A repeat, while reads mapping to MDV075.2 contained 3–31 copies (Fig 2C). For both loci, the most frequent (i.e., consensus) copy number was 10, which was represented by 40 reads (11.3%) in MDV006.5 and 30 reads (9.1%) in MDV075.2. We found no sequencing reads of CVI988-HiFi containing 132B repeats. Additionally, no reads with partial copies of the repeating unit were identified. A Mann-Whitney U test showed a significant difference in 132-bp repeat copy numbers between MDV006.5 and MDV075.2 (U (1) = 53129, p = 0.0408). Out of the three virulent strains, only vv+ strain 675A exhibited reads with copy number variation relative to the Illumina reference, albeit at lower proportions (MDV006.5 = 2 out of 7 (28.6%) reads, MDV075.2 = 6 out of 27 (22.6%) reads). PacBio HiFi reads of MDV strain 675A mapping to both loci exhibited copy numbers ranging from 2-3, but always contained at least one copy of the 132B repeat (Fig 2D). The majority genotype of strain 675A consisted of one 132A repeat followed by one 132B repeat (MDV006.5 = 62.5%, MDV075.2 = 77.2%), and was also shared across both loci. Conversely, a genotype consisting of two copies of the 132B repeat and no 132A repeats was only found in reads mapping to MDV006.5 (14.3%). Similarly, a genotype consisting of one 132A repeat followed by two 132B repeats was only found in reads mapping to MDV075.2 (3.9%). Additionally, we found no reads with more than two copies of the 132B repeat.

## The proline-rich region of MDV049/UL36 is highly variable across MDV strains and involves six alternative versions of a 6-AA repeating unit

The MDV049/UL36 gene encodes a large tegument protein (~3300-AA) that contains a proline-rich region (UL36-PRR) near its carboxyl terminus [15]. This locus, which constitutes ~6% of the MDV049/UL36 coding sequence, harbors the only set of tandem repeats in the MDV genome that are found outside of the structural repeat regions (Fig 1D). Due to the presence of silent mutations at the nucleotide level, the repetitive patterns of the UL36-PRR are typically described in terms of their amino-acid composition (Fig 3A) [43]. The general pattern at this locus consists of 6-AA repeated motifs interrupted at irregular intervals by 10-AA motifs. High-fidelity long read sequencing of the four MDV strains revealed six different versions of the 6-AA repeating unit: "6A" (SPAPKP), "6B" (SPASKP), "6C" (TPAPKP), "6D" (PPAPKP), "6E" (PPASKP), and "6F" (TPASKP) (Fig 3B). We also found three different versions of the 10-AA interrupting motif: "10A" (KPPPDPDFKP), "10B" (KPPPAPDSKP), and "10C" (KPPPTPDSKP). At the consensus level, the versions and copy numbers of 6-AA repeats occurring between the 10-AA motifs were found to be highly variable across all four strains, with CVI988-HiFi being the only one lacking 6B-type repeats and containing 6D- and 6E-type repeats (Fig 3B). For CVI988-HiFi, a total of 640 reads were found to completely contain the UL36-PRR (Table A in S1 File). Of these, 638 (99.7%) showed copy number variation relative to the Illumina reference, specifically in terms of the number of 6A-type repeats occurring between the third and fourth 10AA motifs (4–22 copies) (Fig 3B, blue histogram). For MDV strain Md5-HiFi, only a single read (out of a total of eight) mapping to this region showed evidence of variation relative to the reference. This read was found to contain five copies of the 6C-type repeat prior to the first 10-AA motif, as opposed to the six copies of the 6C-type repeat displayed by the reference sequence. MDV strain 675A had 26 reads mapping to this region, of which a total of 3 (11.5%) showed indels relative to the reference. Two of these reads (7.7%) showed a large 240-bp deletion, resulting in a repetitive pattern consisting of a single 10A motif along with two sets of 6-AA repeats (i.e., Variant 1) (Fig 3B). The remaining read showed a 6F repeat inserted into the first set of 6-AA repeats (i.e., Variant 2). MDV strain HPRS-B14 showed no intrapopulation diversity associated with the UL36-PRR in a total of 43 reads mapping to this locus. To add context to the extent of sequence diversity we found at this locus, we examined the 39 published consensus genomes of MDV used in earlier phylogenetic analyses, which were all sequenced using either Illumina- or Sanger-based

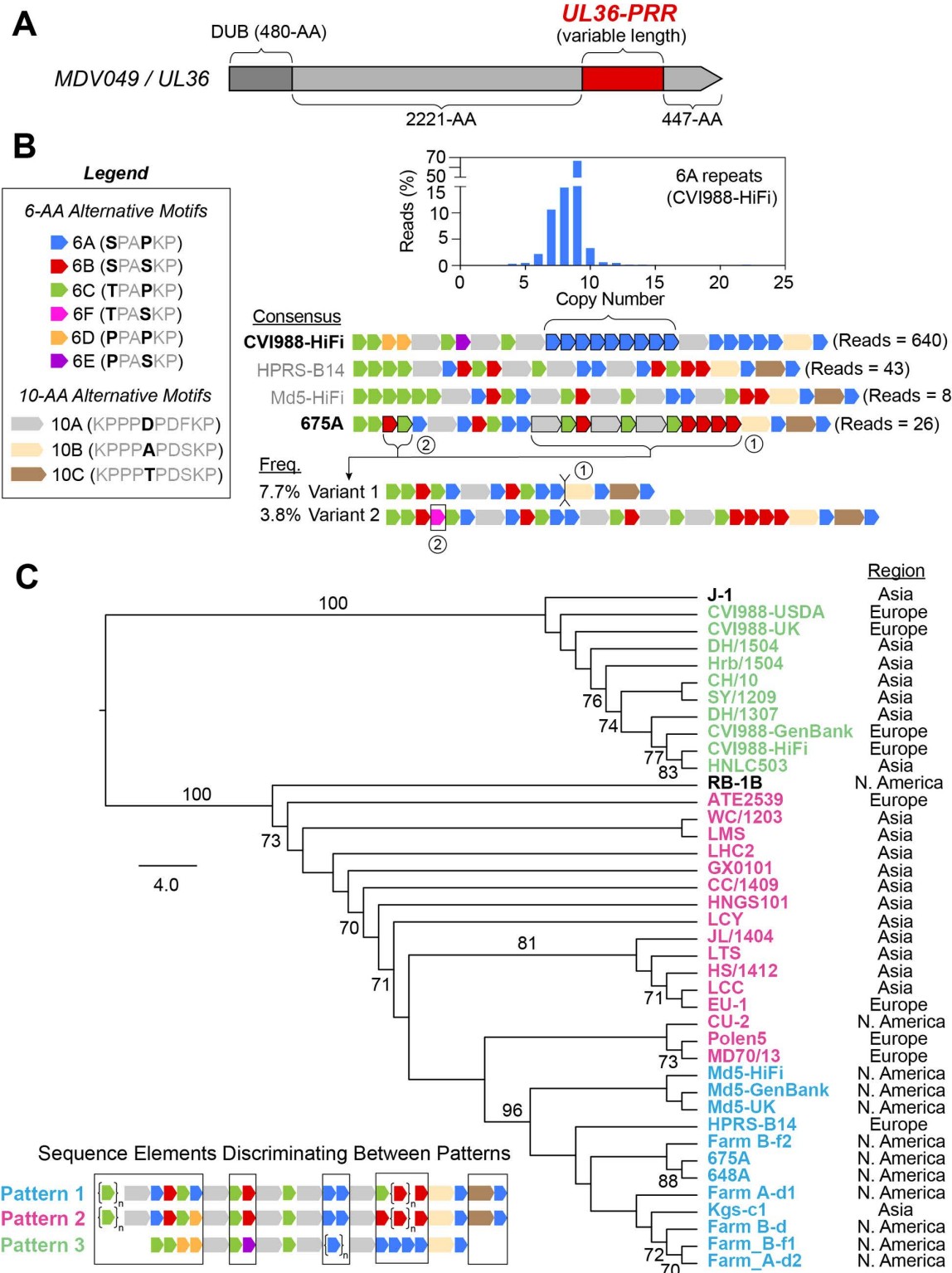

**Fig 3. The proline-rich region of MDV049/UL36 shows distinct patterns of variation in CVI998/Rispens and virulent strains of Eurasian and North American origin. A)** The MDV049/UL36 gene is ~10-kb in length and encodes a large tegument protein (~3300-AA). The N-terminus corresponds to the viral deubiquitinase (DUB) domain. A proline-rich repetitive region (UL36-PRR, dark red) is found near the C-terminus. **B)** The proline-rich

region of MDV049/UL36 exhibits complex repetitive patterns that are typically described at the amino-acid level. The legend depicts the repeating units that occur at this locus, which include six different versions of a 6-AA tandem repeat (named 6A-6E) and three different versions of a 10-AA non-tandem repeat (named 10A-C). The consensus sequence of the MDV049/UL36 proline-rich region is shown for CVI988-HiFi, HPRS-B14, Md5-HiFi, and 675A, with the number of reads for each shown at the right. Strains showing intrapopulation diversity at this locus (bold) included CVI988-HiFi and 675A. CVI988-HiFi variants differed only in terms of the number of copies of 6A repeats in a single segment (bracket and histogram), with copy numbers ranging from 4-22 copies and a modal copy number of 9 copies. PacBio HiFi reads of MDV strain 675A contained 2 minor variants, which either lacked a large segment of repeats (Variant 1, 7.7%) or included a single copy of the 6F repeat between an 6B and 6C repeats in the first set of tandem repeats (Variant 2, 3.8%) **C)** Phylogenetic analysis of the MDV049/UL36 PRR for 40 MDV strains. These include the four strains sequenced using PacBio HiFi and 36 additional MDV strains with published consensus genomes (Table B in S1 File). A maximum-likelihood (ML) tree based on the PRR amino-acid sequence of the 40 strains was generated using the HIVb+F substitution model, with bootstrap values ≥70% shown. The tree is rooted at the midpoint. The lower-left inset shows a summary sequence of the three distinct repetitive patterns identified, with strain names color-coded in the ML tree on the right (Pattern 1 = blue, Pattern 2 = pink, Pattern 3 = green) (see Fig B in S1 File for a more detailed view of PRR alignment). Sequence elements that discriminate between patterns are highlighted in the inset diagram (black boxes). Strains lacking one or more pattern-defining sequence elements are shown in black (i.e., RB-1B, J-1). The geographic origin of each strain is listed to its right.

methods (Fig B and Table B in S1 File). Published consensus genomes of MDV strains Md5-USDA, 814 and HC/0803 were found to lack the UL36-PRR and were excluded from further analyses. Visual inspections of the amino-acid sequence of the remaining 40 proline-rich regions revealed three distinct repetitive patterns (Fig 3C). Pattern 1 (blue) was characterized by a lack of 6D repeats, and the presence of a 6C repeat in the final stretch of 6-AA repeats (Fig B in S1 File). This pattern was highly represented in North American strains, but was also found in HPRS-B14 (Europe) and Kgs-c1 (Asia). Pattern 2 (pink) was characterized by the presence of a single 6D repeat in the second stretch of 6-AA repeats, and by a lack 6C repeats in the final stretch (Fig B in S1 File). This pattern was highly represented in Eurasian strains, but was also found in CU-2 (North America). Pattern 3 (green) was characterized by the lack of 6B repeats, and the presence of the 6E repeat (Fig B in S1 File). This pattern was found only found in CVI988/Rispens genomes (CVI988-GenBank, CVI988-UK, CVI988-USDA, CVI988-HiFi) and in six Chinese strains (CH/10, DH/1307, DH/1504, HNLC503, Hrb/1504, SY/1209) previously described as natural recombinants of CVI988/Rispens. Analysis of the previously published genomes also revealed two additional versions of the 10-AA interrupting motif: "10D" (KPPPAPDFKP) and "10E" (KPPPDPDSKP), for a total of five versions (Fig B in S1 File). Strains RB-1B (North America) and J-1 (Asia) were found to exhibit sequence elements that did not strictly match any of these three repetitive patterns. Phylogenetic analyses of the UL36-PRR consensus amino-acid sequence for all 40 strains revealed that RB-1B clustered near Eurasian strains, while J-1 clustered near CVI988/Rispens (Fig 3C).

### MDV strain CVI988-HiFi variants completely lack the LAT promoter region and exhibit deletions of variable sizes in the 5' end of LAT

MDV genomes contain a family of non-protein-coding transcripts that are overexpressed during latency, collectively known as the latency-associated transcripts (LATs) [44]. These transcripts are all derived from an ~ 11-kb region known as the LAT gene, with the different transcripts being the product of alternative-splicing events. Due to its size, the LAT gene overlaps several MDV open-reading frames (ORFs), including the major transcriptional regulatory protein MDV084/ICP4, in the antisense direction [45]. The 5' end of LAT is associated with a set of 60-bp tandem repeats that act as the LAT promoter (Fig 4A). These repeats are also known as the "p53" repeats due to containing a highly-conserved p53-binding site, and at least two copies are needed for strong promoter activity [46,47]. However, MDV strains with mild and attenuated pathotypes have been shown to sometimes lack any copies of the 60-bp repeating unit and/or contain deletions of variable lengths in the 5' end of LAT (Fig 4A) [48]. A total of 29 MDV variants with 5'LAT deletions, also known as "subtypes", have been identified in past studies (Table C in S1 File) [15,49]. As an alternative to the prior description by Stik et al.[46], a new characterization of the 60-bp repeated motif revealed that these deletions impact the first 32–1,400 nucleotides immediately downstream from the point of overlap with LAT, starting with nucleotide +37 relative to the LAT transcription

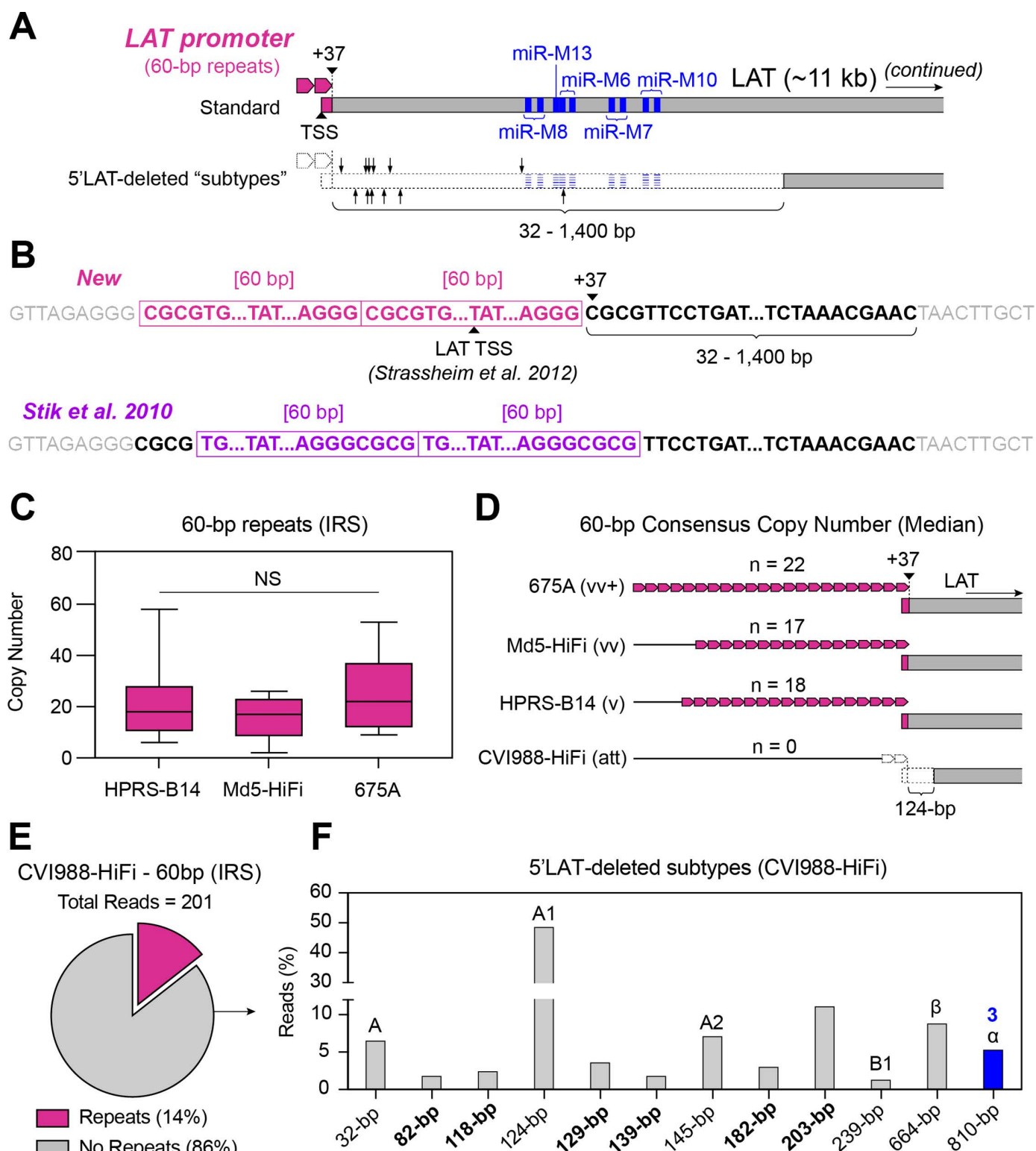

**Fig 4. The absence of repeats in the LAT promoter region of MDV is associated with deletions of varying sizes in 5' end of LAT. A)** The promoter region of the MDV LAT gene consists of 60-bp simple repeats (rose, top), with at least two copies being required for promoter activity. An earlier study by Strassheim et al. proposed that the LAT transcription start site (TSS) is found inside the last 60-bp repeat [50]. However, published genomes of attenuated and mild strains of MDV can lack the 60-bp repeats and exhibit deletions of variable lengths in the 5' end of LAT (i.e., 5'LAT-deleted

"subtypes", bottom). Larger deletions can also impact a cluster of miRNAs (miR-M8-M10, blue). Black arrows indicate the 3' end of the 5'LAT-deleted subtypes found in the CVI988-HiFi sample. **B)** As part of this study, we propose a new characterization of the 60-bp repeated motif (top, rose). Under this model, 5'LAT deletions impact the first 32-1,400 nucleotides immediately downstream from the region of overlap (black font, bold), starting at nucleotide +37 relative to the TSS proposed by Strassheim et al [50]. In a prior description by Stik et al. (bottom, violet) [46], the first 4 nucleotides impacted by 5'LAT deletions were proposed to belong to the non-repetitive region immediately upstream from the 60-bp repeats. **C)** More virulent strains did not show any reads with deletions in 5'LAT, instead harboring long tracts of 60-bp repeats with variable copy numbers, as shown in these box plots (HPRS-B14 = 13 reads, Md5-HiFi = 5 reads, 675A = 5 reads). No significant differences in 60-bp repeat copy numbers were found based on a Kruskal-Wallis test. **D)** Median 60-bp motif copy numbers were used to determine the consensus sequence for all three non-attenuated strains (HPRS-B14 = 18 copies, Md5-HiFi = 17 copies, 675A = 22 copies). The CVI988-HiFi majority variant lacked any copies of the 60-bp repeats and showed a 124-bp deletion in 5'LAT corresponding to the A1 molecular subtype. **E)** CVI988-HiFi reads that still contained the 60-bp repeats (~14%, cerise) had a median copy number of 9 repeats. **F)** A total of twelve 5'LAT-deleted molecular subtypes were identified in the CVI988-HiFi sample, including 6 novel subtypes (indicated in bold). Only one of these subtypes (810-bp deletion, blue) impacted the cluster of miRNAs further downstream, with a total of three miRNAs (miR-M8, miR-M13, miR-M6) being impacted.

start site (TSS) proposed by Strassheim et al. (Fig 4B) [50]. Larger deletions (>683-bp) start to impact a cluster of miRNAs encoded by LAT, which have been shown to play a role in MDV pathogenesis [50,51]. PacBio HiFi reads of HPRS-B14, Md5-HiFi, and 675A mapping to the 60-bp repeats located near the IRL/IRS junction (HPRS-B14 = 13 reads, Md5-HiFi = 5 reads, 675A = 5 reads; Table A in S1 File) all showed a range of different 60-bp repeat copy numbers (HPRS-B14 = 6–58 copies, Md5-HiFi = 2–26 copies, 675A = 9–53 copies) (Fig 4B). Since no two reads were found to contain the same number of copies of the 60-bp repeating unit, the median copy number was used to determine the consensus sequence for each of these three strains (HPRS-B14 = 18 copies, Md5-HiFi = 17 copies, 675A = 22 copies) (Fig 4C). A Kruskal-Wallis H test showed no significant differences in 60-bp repeat copy numbers (H (2) = 0.4947, p = 0.7809). For CVI988-HiFi, a total of 201 reads were found mapping to the LAT promoter region, of which 172 (85.6%) completely lacked the 60-bp repeats (Fig 4D, 4E). Among reads lacking repeats, 83 (48.3%) exhibited a 124-bp deletion past the point of overlap, corresponding to the A1 subtype (Fig 4F, Table C in S1 File). We also found reads exhibiting the α subtype (5.2%), the β subtype (8.7%), the A subtype (6.4%), the A2 subtype (7%), and the B1 subtype (1.2%). A subset of reads (34.3%) showed deletions extending past the point of overlap that did not correspond with any known subtypes, including reads with a 203-bp deletion (11%), a 182-bp deletion (2.9%), a 139-bp deletion (1.7%), a 129-bp deletion (3.5%), a 118-bp deletion (2.3%), and an 82-bp deletion (1.7%) (Fig 4F, Table C in S1 File). A small minority of CVI988-HiFi reads (14.4%) did contain copies of the 60-bp repeating unit (Fig 4E). For these reads, 60-bp repeat copy numbers ranged from 0-25 copies (median = 9). None of the more virulent MDV strains (HPRS-B14, Md5-HiFi, 675A) showed any signs of LAT deletions in the sequencing reads available.

### The length and composition of the multiple telomeric repeats (mTMR) region of the MDV *a*-like sequence is highly variable across strains

The MDV *a*-like sequence occurs at the genomic termini and the IRL/IRS junction, where it is reported to play a central role in replication, pathogenesis and tumor formation [52]. The name "*a*-like" is derived from early comparisons to the HSV-1 *a* sequence, which exhibits a similar structure and is located in the same genomic sites in the HSV-1 genome [53]. However, a distinct feature of the MDV *a*-like sequence is the presence of two sets of 6-bp "telomeric" repeats. The sequence of these repeats (TTAGGG) is identical to the chicken host telomeric sequences, enabling the virus to integrate into the host telomeres via homologous recombination during latency [54,55]. The first set of repeats, known as the short telomeric repeats (sTMR), has been observed to always contain exactly six copies of the 6-bp repeating unit (Fig 5A) [56]. In contrast, the second set has been shown to be highly variable in copy number, and is known as the multiple telomeric repeats (mTMR) [57]. The telomeric repeats occurring in the mTMR are also interrupted at varying intervals by a recurring 13-bp motif (TTCAGGCCTAGGG) (Fig 5A–5B). Previous studies have described this repetitive pattern as "stretches" of 6-bp telomeric repeats interrupted by 13-bp "islands" [15,21]. In addition to the sTMR and the mTMR, the MDV a-like

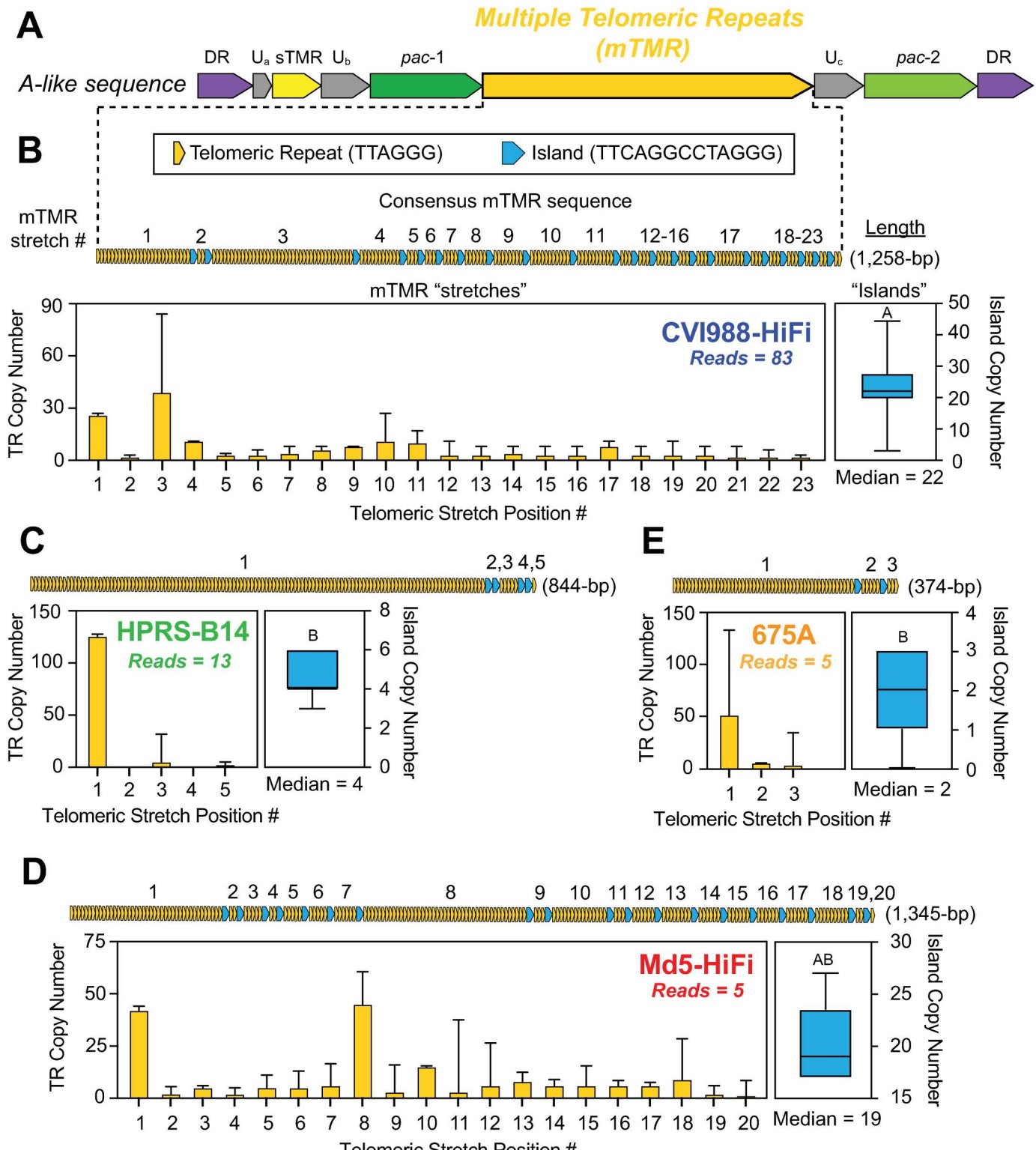

**Fig 5. The mTMR region of the MDV *a*-like sequence shows extensive intrapopulation diversity in all four strains sequenced using PacBio HiFi. A)** The *a*-like sequence of MDV, which occurs in the IRL/IRS junction and the genomic termini, begins and ends with a 12-bp motif (GGCCGCGA-GAGG) known as direct repeat (DR) (violet). The initial DR is followed by six copies of a 6-bp motif (TTAGGG), collectively known as the short telomeric

repeats (sTMR) (yellow). Near the center of the *a*-like sequence lies another region harboring 6-bp telomeric repeats, which varies in length and is therefore known as the multiple telomeric repeats (mTMR) (dark yellow). Telomeric repeats in the mTMR are interrupted at irregular intervals by "islands" consisting of a 13-bp motif (TTCAGGCCTAGGG; blue). The remainder of the MDV *a*-like sequence consists of the *pac*-1 (dark green) and *pac*-2 (green) motifs, in addition to three small segments containing quasi-unique sequences (U$_a$, U$_b$ and U$_c$) (grey). **B-E)** To resolve the mTMR length and repeat composition, we calculated the median number of copies of telomeric repeats at each repetitive stretch (bar plots, dark yellow) and the median number of interrupting islands (box plots, blue). Error bars represent the standard deviation for the copy number of telomeric repeats in each stretch (yellow bars) or the total number of islands (blue boxes). Letters above island box plots represent statistically significant differences across strains based on pairwise comparisons using Dunn's test (Table D in S1 File). A graphical representation of the resulting consensus sequence is shown for each stretch and for the consensus-level mTMR of each strain as a whole, followed by the length of the consensus sequence.

sequence also contains several non-repeating motifs. These include the packaging signals *pac*-1 and *pac*-2, as well as two direct repeats (DR), which mediate cleavage of MDV genomes from concatemeric intermediates generated during rolling-circle replication [58]. For all four strains, PacBio HiFi reads mapping to the *a*-like sequence were always found to extend into the nearby LAT promoter region, resulting in both features co-occurring within the same reads (Table A in S1 File). To account for the presence and diversity of 5'LAT-deleted molecular subtypes in CVI988-HiFi, for this strain, only reads harboring the majority A1 subtype (83 out of 201 reads) were inspected (Tables A and C in S1 File). In the sTMR, reads of CVI988-HiFi, HPRS-B14, Md5-HiFi and 675A all showed exactly six telomeric repeat copies, suggesting that this locus does not exhibit intrapopulation diversity. In the mTMR, reads of all four strains showed extensive differences in the total number of 13-bp islands and in telomeric repeat copy numbers at each stretch (Fig 5B–5E). The consensus mTMR of each of the four strains was determined by first calculating the median number of islands (CVI988-HiFi = 22 islands, HPRS-B14 = 4 islands, Md5-HiFi = 19 islands, 675A = 2 islands) (Fig 5B–5E, blue histograms). Pairwise comparisons of island copy numbers using Dunn's test showed a significant difference between CVI988-HiFi and non-attenuated strains HPRS-B14 and 675A (Table D in S1 File). The total number of stretches in the consensus for each strain was always one more than the number of islands, with every island flanked by two stretches. We then graphed the median copy number of telomeric repeats at each stretch (Fig 5B-5E). In addition to copy number variation in the mTMR, a subset of reads from each of the four strains exhibited *a*-like sequences with atypical structures (Fig C in S1 File). These included: reads with reiterations of the *pac-1* and *pac-2* motifs accompanied by additional instances of the mTMR region, reads with extra copies of the entire *a*-like sequence, and reads with combinations of both. Reads with 3 copies of the entire *a*-like sequence were only found for HPRS-B14, while reads showing duplications of the *pac-2* motif were only found for CVI988-HiFi (Fig C in S1 File).

### The proline-rich region of the Meq oncoprotein involves three distinct amino-acid repeating units

The *meq* oncogene (MDV005/MDV076) encodes the major MDV oncoprotein Meq, which has been shown to be indispensable for MDV-induced transformation of T lymphocytes [59,60]. Variation in Meq has been historically described in terms of "isoforms", with the standard "Meq" being 339-AA in length [61]. "Long Meq" (L-Meq) has been described as a longer version of Meq containing a 59–60-AA insertion in the transactivation domain (TAD) [62]. "Short Meq" (S-Meq) has been described as a shorter version of Meq exhibiting a 41-AA deletion in the TAD [63]. Alternatively, "Very Long Meq" (VL-Meq) and "Very Short Meq" (VS-Meq) isoforms have only been described recently, with the first examples showing a 79-AA insertion and 74-AA deletion in the TAD, respectively [64,65]. Similar to MDV049/UL36, the C-terminus of Meq contains a region that is known to harbor proline-rich repeats (Meq-PRR) (Fig 6A). Past descriptions of these repeats by Jones et al., Lee et al., and Chang et al. have suggested that they start at or around AA residue 149 [62,66–69]. However, these early characterizations disagree on the length and the number of repeats associated with each Meq "isoform". To account for these inconsistencies, we performed manual repeat decompositions of the Meq-PRR based on a multiple-sequence alignment of PacBio HiFi consensus genomes. Three distinct repeating units were identified, which were 27-AA, 14-AA, and 19-AA in length, respectively, and started at AA residue 169 (Fig 6A–6C). Additionally, more than

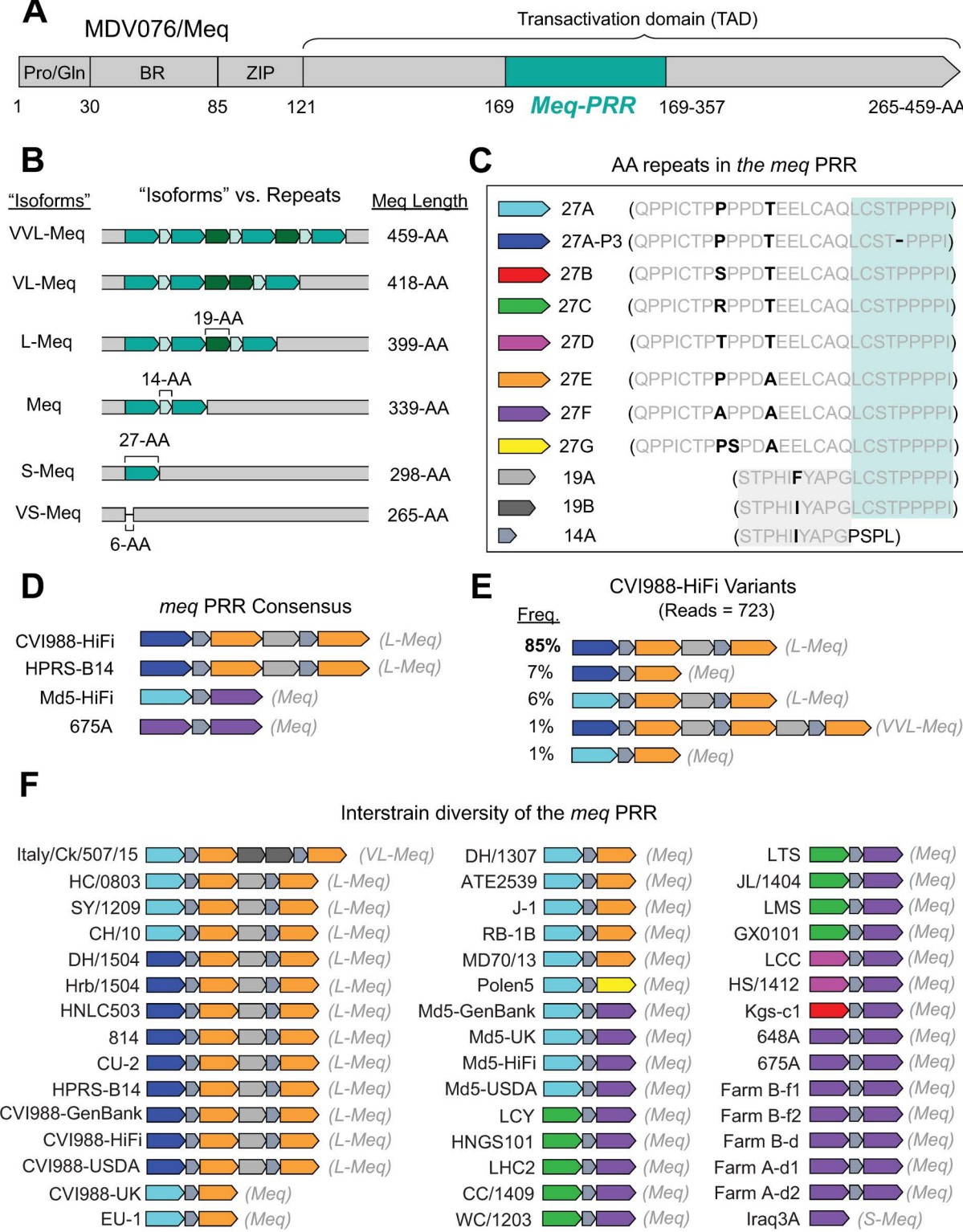

**Fig 6. Manual repeat decompositions reveal three distinct repeating units in the proline-rich region of Meq. A)** The Meq oncoprotein is comprised of an N-terminal proline/glutamine (Pro/Gln) rich domain, a basic region (BR), a leucine zipper (ZIP) domain and a transactivation domain (TAD). The TAD contains a proline-rich region (PRR, aqua) associated with a major repetitive element that is typically described at the amino-acid level. **B)** The

prior "isoform" model of variation in Meq can alternatively be represented as differences in the number of copies of three newly identified repeating units. **C)** The newly identified repeating motifs which make up the Meq-PRR include: eight alternative versions of a 27-AA repeat, two alternative versions of a 19-AA repeat, and one 14-AA repeat. All three repeating units bear sections of sequence commonality (gray vs. black letters, gray and blue shading). **D)** Consensus sequence of the Meq-PRR for the four strains sequenced using PacBio HiFi reads. Meq-PRRs are depicted with color-coding to indicate the identity and copy number of each repeating unit, with the corresponding isoform-based description indicated to the right (grey text). **E)** Relative frequencies of CVI988-HiFi variants identified through visual inspection and manual re-alignment of reads mapping to MDV076/Meq. **F)** Comparison of the Meq-PRR across 43 MDV strains, including the present 4 determined by PacBio HiFi (bold) and 39 previously published Meq-PRRs (see Tables B and E in S1 File for accessions).

one version of the 27-AA repeating unit was identified (Fig 6C). PacBio HiFi reads of non-attenuated strains mapping to MDV076 (HPRS-B14 = 15 reads, Md5 = 2 reads, 675A = 36 reads; Table A in S1 File) were not found to contain indels in the TAD (Fig 6D). Visual inspection of CVI988-HiFi reads mapping to this locus (n = 734) revealed a variant with a 120-AA insertion in the TAD (Fig 6E). This variant did not match any of the known isoforms and was provisionally labeled as "Very, Very Long Meq" (VVL-Meq) (Fig 6B, 6E). To test the ability of the newly identified repeating units to reliably account for structural variation in the Meq-PRR, we extended our analyses to the Meq sequences of the 39 MDV strains used in earlier phylogenetic analyses, as well as to 3 additional strains representing the VL-Meq (Italy/Ck/507/15), VS-Meq (Ck/IR/99–35/2021), and S-Meq (Iraq3A) isoforms (Table E in S1 File). A multiple-sequence alignment of these Meq-PRRs confirmed that the three repeating units could reliably account for structural differences across all known Meq isoforms (Fig 6B). In this model, standard Meq contains two 27-AA repeats, interrupted by a 14-AA motif. Longer forms such as L-Meq, VL-Meq, and VVL-Meq extend this pattern via a new 19-AA repeat, plus additional copies of the 14-AA and 27-AA repeats. In contrast, shorter forms such as S-Meq and VS-Meq involve sequential loss of these repeats, with total repeat loss in VS-Meq also accompanied by deletions in the non-repetitive region immediately downstream (Fig 6B). A total of eight different versions of the 27-AA motif (27A, 27A-P3, 27B, 27C, 27D, 27E, 27F, 27G), a single version of the 14-AA motif (14A), and two different versions of the 19-AA motif (19A, 19B) were identified across the 44 Meq-PRRs available for analysis (Fig 6C, 6F). At the consensus level, the repetitive patterns of CVI988-HiFi and HPRS-B14 were found to match each other in length and composition (Fig 6D). These two strains were also found to not share any versions of the 27-AA repeating unit with Md5-HiFi and 675A, which both contained 27F repeats. Out of the four strains, only CVI988-HiFi was observed to exhibit intrapopulation diversity at this locus (Fig 6E).

### Detection of CVI988-HiFi variants with a 4.3-kb deletion in the Unique Short region

A subset of 79 (20%) CVI988-HiFi reads mapping to the IRS/US junction were found to exhibit an atypical US region (Fig D in S1 File). The first 186 nucleotides of this atypical US region corresponded to the reverse complement of the last 186 nucleotides of a typical US region (Fig D in S1 File). The next 31 nucleotides were not found to map anywhere else in the MDV genome. These were immediately followed by a 4.3 kb deletion that removed SORF1, SORF2, US1, US10, SORF3, and approximately half of US2. PacBio HiFi reads mapping to this locus showed a variety of start and end points, suggesting that these variants were part of otherwise intact genomes (Fig D in S1 File).

### Discussion

Identifying genomic regions that contribute to observed phenotypic differences across Marek's disease virus (MDV) strains remains of high importance to the field. However, the limitations of DNA sequencing technologies have made it difficult to account for genomic diversity in MDV loci harboring long stretches of tandem repeats [70,71]. To address this gap in knowledge, we sequenced four MDV strains using a PacBio high-fidelity (HiFi) long read platform: CVI988-HiFi (attenuated), HPRS-B14 (virulent), Md5-HiFi (very virulent), and 675A (very virulent plus) [13]. In addition to showcasing the ability of PacBio HiFi reads to resolve complex tandem repeat patterns in the MDV genome, our results suggest that all of these loci are highly polymorphic and exhibit patterns of variation that may be contributing to phenotypic differences

across strains. The forces that generate heterogeneity at these repeats and shape their evolution over time are an area for future exploration.

To date, only a handful of studies have used long-read sequencing approaches to generate herpesvirus consensus genomes [72–78]. However none of these studies – including those with high-fidelity long sequence reads – have analyzed variation in tandem repeat regions. Past studies that have tackled this challenge have either relied on Sanger sequencing of cloned or PCR fragments, or on the estimation of tandem repeat copy numbers from Illumina sequencing coverage depth [79–82]. While these studies provided an important foundation for the present work, prior methods were limited in their ability to identify novel versions of repeating motifs, or to assess the full extent of intra-population tandem repeat diversity. By accurately resolving the complex patterns of MDV's repeat regions, we demonstrate both the rationale and a viable approach for future application to other herpesviruses and DNA-based pathogens.

The decision to perform multiple steps of our analysis manually as opposed to relying on widely-used bioinformatic tools stems from the lack of benchmarking studies performed in the context of large DNA viruses like MDV [83–85]. Here, we demonstrate that avian herpesviruses can exhibit high levels of intrapopulation diversity in tandem repeats [82,86], a phenomenon that uniquely combines the two biggest challenges faced by modern bioinformatic algorithms [87,88]. We also show that MDV tandem repeats can exhibit complex repetitive patterns involving multiple repeating units and/or alternative versions of a repeating unit. Since most widely-used bioinformatic tools have not been designed with these challenges in mind, our approach, although limited in scalability, offers a more reliable outcome and provides a ground truth for future benchmarking studies [89]. Likewise, while the relatively low sequencing coverages obtained for HPRS-B14, Md5-HiFi, and 675A limited our ability to comprehensively assess the genomic diversity of these samples, our use of a sequencing protocol that did not include any amplification steps enabled us to provide a similar foundation for future studies seeking to adapt target enrichment protocols for MDV and other herpesviruses [88,90].

Consistent with prior observations, we found that the heavily passaged and attenuated vaccine strain CVI988-HiFi exhibits extensive intrapopulation diversity in the 132-bp tandem repeats overlapping the MDV006.5/MDV075.2 transcripts [91–93]. More specifically, CVI988-HiFi reads fully containing the 132-bp repeats displayed a wide range of repeat copy numbers (2–37 copies), and the consensus (i.e., most frequent) copy number in each locus accounted for only 9–12% of the reads. Across all four strains in this study, only MDV strain 675A (vv+) contained any copies of the alternative 132-bp motif with a C>T base change in nucleotide 67 (132B). All of the viral genomes in the 675A sample contained at least one copy of the 132B repeat, and no other strains harbored any copies of this variant. Since MDV strains used in past serial passage experiments (e.g., Md5, Md11) only contained the 132A version of the repeating unit, the biological relevance of the 132B variant remains largely unexplored [41,42,92,94,95]. Our findings support earlier observations by Spatz et al. suggesting a potential link between the presence of the 132B repeat and the vv+ pathotype [15]. Notably, since both versions of the 132-bp repeat share the same amino-acid sequence, any phenotypic effects resulting from the C>T base change are likely stemming from impacts at the nucleic acid level (e.g., secondary structures and/or changes in translation efficiency).

The proline-rich region of MDV049/UL36 (UL36-PRR) has been proposed as the single most divergent locus between CVI988/Rispens and virulent strains [14,15]. PRRs are known to have special signaling and protein-protein interaction functions, and the presence of a PRR in the C-terminus of UL36 is conserved across all herpesviruses [96–98]. Nevertheless, functional studies of MDV049/UL36 have primarily focused on the N-terminus, which contains the viral deubiquitinase (DUB) catalytic domain [99,100]. In this study, we identified multiple new repetitive motifs in UL36-PRR [15,101], including six versions of a 6-AA tandem repeat and five versions of a 10-AA interrupting motif (Fig 3 and Fig B in S1 File). We found that PRR patterns at this locus appear to segregate with the geographic origin of non-attenuated MDV strains. In addition, we found a unique pattern in the CVI988/Rispens vaccine strain, which is reflected in a set of 6 Chinese isolates that have been described as field-based vaccine recombinants [102–104]. Notably, the intrapopulation diversity of the UL36-PRR was limited to specific repeat segments, suggesting that these motifs could be used as reliable biomarkers.

This UL36 region could even offer alternative qPCR targets to the commonly used "SNP #320" of pp38 for differential quantification of CVI988/Rispens and virulent strains in field samples [31].

In addition to our findings for the UL36-PRR, we also investigated the repetitive patterns associated with the proline-rich region of Meq (Meq-PRR), the major MDV oncoprotein. Contrary to past descriptions of this locus, we have identified three distinct repeating units (of 14, 19, and 27 AA in length) in the Meq-PRR, which were mostly found to reiterate in an alternating pattern (Fig 6A). Although accurate decomposition of complex tandem repeats remains a significant challenge, the three proposed repeating units can reliably account for all of the structural differences observed across known Meq "isoforms" (Fig 6F) [105]. Notably, our new model of the Meq-PRR highlights a number of similarities between this locus and the UL36-PRR (Fig 3B and Fig B in S1 File), with both proline-rich regions exhibiting complex repetitive patterns that are highly variable across MDV strains [15]. In contrast, past functional studies of the Meq-PRR repeats have largely been based on the model proposed by Jones et al. [66], which characterized Meq as a simple repeat with different boundaries [61,67,101,106,107]. Future studies could explore the functional significance of both of these proline-rich repeat regions (i.e., in Meq and UL36) and the mechanisms contributing to the emergence of these complex repetitive patterns.

In contrast to the complex repetitive patterns found in proline-rich regions, we found no evidence of alternative motifs or additional repeating units associated with the 60-bp repeats that make up the MDV LAT promoter. Past experiments by Labaille et al. identified 29 molecular "subtypes" of 5'LAT deletions across six commercial batches of the CVI988/Rispens vaccine [49]. We confirmed the presence of six of these subtypes in the CVI988-HiFi sample (A1, α, β, A, A2 and B1) and identified six novel subtypes with deletions sizes ranging from 82-203-bp, relative to the LAT TSS [50]. We also found that a small subset (~14%) of CVI988-HiFi reads still retained a functional LAT promoter (i.e., at least 2 copies of the 60-bp repeat). This discovery may help account for past reports of CVI988/Rispens vaccine batches showing LAT expression despite largely being composed of 5'LAT-deleted subtypes [108,109]. While these findings provide the most complete picture of 5'LAT diversity to date, the molecular mechanisms giving rise to 5'LAT-deleted subtypes are still largely unknown. Interestingly, a similar pattern of variation was observed in the Meq-PRR for MDV strain Ck/IR/99–35/2021 (VS-Meq) [65]. In this strain, the absence of proline-rich repeats was also accompanied by an 18-bp deletion in the non-repetitive region immediately downstream. Future studies are warranted to explore the relationship between repeat loss and deletions in adjacent non-repetitive regions, and determine whether Meq-PRR deletions are driven by the same molecular mechanisms that lead to 5'LAT-deleted subtypes. In vivo studies will be critical to understand the impact of 5'LAT deletions on attenuation and tumor formation.

Similar to the MDV006.5/MDV075.2 transcripts, the mTMR region of the MDV a-like sequence has been shown to expand during serial passage in cell culture [93]. In contrast, the nearby sTMR region has been described as always consisting of exactly six copies of a 6-bp telomeric repeat (TTAGGG) [53,56]. Our data support these observations, with all four strains showing extensive sequence diversity in the mTMR region, and high conservation in the sTMR region. While the presence of interrupting motifs (e.g., the 13-bp islands) is often associated with reduced repeat expansion and increased stability, our findings suggest that the MDV mTMR region is highly polymorphic and unstable [110]. Further studies will be needed to assess the functional impact of different mTMR repeat compositions, and to determine whether similar levels of intrapopulation diversity can be found in vivo. In addition to copy number variation, we also found structural variants of MDV with 1–2 additional copies of the entire a-like sequence, as well as variants with extra copies of the packaging (pac) motifs and mTMR arrays (Fig C in S1 File). To the best of our knowledge, there is no precedent in the literature for a single a-like sequence containing multiple copies of pac-1, pac-2, or even additional mTMR arrays. Future experiments will be necessary to determine the impact of these variants on viral integration and virulence.

In addition to our assessments of genomic diversity in repetitive loci, we detected variants of CVI988-HiFi with a 4.3-kb deletion in the US region. Although naturally-occurring MDV variants with deletions of this magnitude have not been reported in prior studies, two deletion mutants constructed by Parcells et al. from MDV strains GA (GAΔ4.8lac) and RB-1B (RB-1BΔ4.5lac) showed remarkably similar deletions (4.8-kb and 4.5-kb, respectively) in the US region [111,112]. These

deletion mutants also lacked SORF1, SORF2, US1, US10, SORF3, and US2, but had the *lacZ* gene of *Escherichia coli* inserted at the site of the deletion. Experimental assays conducted by the authors showed that the loss of the six genes did not completely remove the ability of MDV to grow in cell culture or to induce tumors in live birds, although the recombinants were impaired in both outcomes compared to wild-type [111,112]. In this context, our data suggest that studies of CVI988/Rispens should check for similar deletion variants, to ascertain their potential contribution to its attenuation. Alternatively, the detected variants could represent defective viral genomes (DVGs) [113]. Past reports of DVGs in MDV are limited; however, they are known to occur in herpesviruses during high-multiplicity infections and serial passage, and their presence in other live-attenuated viral vaccines is well-documented [114–119].

Beyond the interstrain and within-sample differences associated with each individual MDV locus harboring tandem repeats, an overarching finding of the present study was that this sample of CVI988/Rispens exhibited unique patterns of variation in almost all of these loci relative to non-attenuated strains (**Fig 1D**). In the 132-bp repeats associated with the MDV006.5/MDV075.2 transcripts, CVI988-HiFi was the only strain to exhibit more than two repeat copies at the consensus level, while also exhibiting the highest copy number diversity at the population level. In the UL36-PRR, CVI988-HiFi lacked any copies of the 6B repeat motif, and uniquely contained a single copy of the 6E alternative motif. CVI988-HiFi was also the only strain to contain any within-sample diversity in the Meq-PRR, which included the first, and to date only, detection of a VVL-Meq "isoform". Likewise, *a*-like sequence variants with an additional *pac*-2 motif, and 5'LAT-deleted subtypes lacking copies of the 60-bp repeats of the LAT promoter, were only found in CVI988-HiFi. The differences add significantly to our understanding of genetic differences between CVI988/Rispens and circulating virulent strains of MDV, since non-repetitive regions of the MDV genome share >98% sequence identity. Future studies will be needed to test for functional consequences of the differences observed at each of these loci, and to more deeply assess additional vaccine and virulent strains of MDV for variability in their repeat regions.

While the above findings significantly advance our understanding of MDV tandem repeats, a number of limiting factors still need to be addressed before long-read approaches can be utilized to their full potential. First, obtaining sufficient quantities of MDV DNA from field samples remains a significant challenge, which is exacerbated by higher DNA input requirements of long-read sequencing platforms (e.g., PacBio) vs. short-read (e.g., Illumina) platforms [88,120,5]. Hybridization-based enrichment protocols that are compatible with long-read sequencing have emerged as a potential way to address this challenge for single-species samples [90], but such protocols have yet to be adapted to viruses in the context of *in vivo* infections. Second, the levels of intrapopulation diversity associated with MDV repeats present unique challenges. The impact of amplification steps during enrichment-based approaches have not yet been investigated for long-read sequencing of repeat regions with intra-population diversity [88,121]. This diversity also presents a challenge for reporting a viral consensus genome. Here we have used the median genotype in place of the consensus; however, a pan-genome approach may prove useful as our knowledge of viral diversity in repeat regions expands [122,123]. Finally, public databases like GenBank do not currently allow for the inclusion of tandem repeat variants when depositing consensus genomes [124]. While this is also true for single-nucleotide variants, tandem repeat variants exhibit much higher levels of intrapopulation diversity. Text files in tabular formats (e.g., TSV) containing information on copy number variation are already routinely submitted as part of data entries for repositories like the Genomic Data Commons (GDC) [125]. Allowing researchers to append such files when submitting viral consensus genomes to GenBank would allow for more accurate representations of viral samples and facilitate large-scale computational analyses of structural variation.

## Supporting information

**S1 File. Fig A: PacBio HiFi consensus genomes cluster near previously published Illumina genomes of the same strain or near strains of the same pathotype.** Using MAFFT, we aligned the consensus genomes of CVI988-HiFi, HPRS-B14, Md5-HiFi, and 675A generated using PacBio HiFi, along with 39 additional consensus genomes belonging to strains from North America, Asia and Europe (see Table B in S1 File for accessions). A maximum-likelihood tree

with gaps excluded was generated using the K3Pu+F + G4 substitution model, with bootstrap values [3]70% shown. The tree is rooted at the midpoint. Arrows indicate newly assembled PacBio HiFi viral consensus genomes. **Fig B: Multiple-sequence alignment of 40 UL36-PRRs reveals three distinct repetitive patterns.** Graphical representation of the multiple sequence alignment used for phylogenetic analysis of the MDV049/UL36 proline-rich region (UL36-PRR) for 40 MDV strains with published consensus genomes (see Table B in S1 File for accessions). Amino acid sequences were initially aligned using MAFFT and then manually curated to improve alignment of the repeating units. The color-coding of strain names relates to the patterns and the tree shown in Fig 3C. Strains exhibiting Pattern 1 have names labelled in blue, Pattern 2 in pink, and Pattern 3 in green. MDV strains sequenced using PacBio HiFi are indicated with an arrow. **Fig C: Structural variants of the a-like sequence harbor partial duplications of the pac-1 and pac-2 motifs. A)** Using PacBio HiFi sequencing, several structural variants of the a-like sequence were identified. Common variants (>20% frequency in at least one strain) included reads with the "standard" 1 copy of the entire a-like sequence, as well as reads with a partial duplication of the a-like sequence involving the pac-1 (dark green), mTMR (dark yellow) and $U_c$ segments (gray) ("Duplicated pac-1"). Rare variants included reads with any of the following: 2 or 3 copies of the entire a-like sequence; a partial duplication involving the pac-2 (green), mTMR and $U_c$ segments ("Duplicated pac-2"); or 2 copies the entire a-like sequence interspaced by a partial duplication involving the pac-2, mTMR and $U_c$ segments ("Complex"). For each variant, duplicated segments are indicated with brackets below and/or above the sequence diagram. **B)** Bar graph showing the relative frequency of a-like sequence structural variants in the PacBio HiFi reads for each of the four strains. Variants present at frequencies >20% (dotted line) for at least one strain were classified as "common" variants, while variants below this threshold across all four strains were classified as "rare". **Fig D: PacBio HiFi reads enable detection of CVI988-HiFi variants with a 4.3-kb deletion in the Unique Short region. A)** Graphical depiction of CVI988-HiFi variant showing an atypical US region with a large deletion and rearrangement. The first 186 nucleotides of this atypical US region correspond to the reverse complement of the last 186 nucleotides of a typical US region. The next 31 nucleotides do not map anywhere in the MDV genome. Thereafter, the reads have a 4.3 kb deletion compared to the majority of CVI988-HiFi reads, which removes SORF1, SORF2, US1, US10, SORF3, and approximately half of US2. **B)** The 79 PacBio HiFi reads supporting this US deletion variant in the CVI988-HiFi sample exhibited a wide range of start- and end-points, suggesting that these variants may be part of otherwise intact genomes. **Table A:** Number of PacBio HiFi reads used to assess intrapopulation diversity at each repetitive locus for all four strains. **Table B:** Published genomes and accessions for additional 39 included strains. **Table C:** List of previously reported and newly identified MDV 5'LAT-deleted molecular subtypes. **Table D:** Dunn's test of mTMR island copy numbers, which relates to Fig 5B-E. **Table E:** Representative strains and accessions of the VL-Meq, S-Meq and VS-Meq isoforms. (PDF)

## Acknowledgments

The authors thank members of the Szpara, Kennedy, and Nair labs for helpful feedback and discussion. The authors would like to acknowledge the support of Craig Paul, Daniel Hannon and other personnel of the Huck Institutes' Genomic Core Facility (RRID:SCR_023645), as well as their PacBio Sequel II platform and other instrumentation.

## Author contributions

**Conceptualization:** Alejandro Ortigas-Vasquez, David A. Kennedy, Moriah L. Szpara.

**Data curation:** Alejandro Ortigas-Vasquez, Daniel W. Renner.

**Formal analysis:** Alejandro Ortigas-Vasquez.

**Funding acquisition:** Yongxiu Yao, Venugopal Nair, David A. Kennedy, Moriah L. Szpara.

**Investigation:** Alejandro Ortigas-Vasquez, Christopher D Bowen, Susan J Baigent.

**Methodology:** Alejandro Ortigas-Vasquez, Christopher D Bowen, Susan J Baigent.

**Project administration:** David A. Kennedy, Moriah L. Szpara.

**Resources:** Susan J Baigent, Yaoyao Zhang, Yongxiu Yao, Venugopal Nair, Moriah L. Szpara.

**Software:** Alejandro Ortigas-Vasquez, Daniel W. Renner.

**Supervision:** Yaoyao Zhang, Yongxiu Yao, Venugopal Nair, David A. Kennedy, Moriah L. Szpara.

**Validation:** Alejandro Ortigas-Vasquez, Christopher D Bowen, Daniel W. Renner.

**Visualization:** Alejandro Ortigas-Vasquez, David A. Kennedy, Moriah L. Szpara.

**Writing – original draft:** Alejandro Ortigas-Vasquez.

**Writing – review & editing:** Alejandro Ortigas-Vasquez, David A. Kennedy, Moriah L. Szpara.

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
