## [Decision Letter · Decision Letter 0]

30 Jun 2025

PPATHOGENS-D-25-01313

High-Fidelity Long-Read Sequencing of an Avian Herpesvirus Reveals Extensive Intrapopulation Diversity in Tandem Repeat Regions

PLOS Pathogens

Dear Dr.Szpara,

Thank you for submitting your manuscript to PLOS Pathogens. After careful consideration, we feel that it has merit but does not fully meet PLOS Pathogens's publication criteria as it currently stands. Therefore, we invite you to submit a revised version of the manuscript that addresses the points raised during the review process.

Please submit your revised manuscript within 30 days. If you will need more time than this to complete your revisions, please reply to this message or contact the journal office at plospathogens@plos.org. Please include the following items when submitting your revised manuscript:

* A rebuttal letter that responds to each point raised by the reviewers (#2 and #3). You should upload this letter as a separate file labeled 'Response to Reviewers'. This file does not need to include responses to any formatting updates and technical items listed in the 'Journal Requirements' section below.

We look forward to receiving your revised manuscript.

Kind regards,

Deepak Shukla

Academic Editor

PLOS Pathogens

Robert Kalejta

Section Editor

PLOS Pathogens

Sumita Bhaduri-McIntosh

Editor-in-Chief

PLOS Pathogens

orcid.org/0000-0003-2946-9497

Michael Malim

Editor-in-Chief

PLOS Pathogens

orcid.org/0000-0002-7699-2064

**Journal Requirements:**

At this stage, the following Authors/Authors require contributions: Alejandro Ortigas-Vasquez, Christopher D Bowen, Daniel W Renner, Susan J Baigent, Yaoyao Zhang, Yongxiu Yao, Venugopal Nair, David A Kennedy, and Moriah L Szpara. Please ensure that the full contributions of each author are acknowledged in the "Add/Edit/Remove Authors" section of our submission form.

https://journals.plos.org/plospathogens/s/submission-guidelines#loc-parts-of-a-submission

4) Please upload a copy of Figure S1-S4 which you refer to in your text on pages 12, 15, 19, and 21. Or, if the figure is no longer to be included as part of the submission please remove all reference to it within the text.

**Reviewers' Comments:**

Reviewer's Responses to Questions

**Part I - Summary**

Reviewer #1: The authors have responded appropriately to the issues raised by the reviewers.

Reviewer #2: In this revised manuscript titled: High-Fidelity Long-Read Sequencing of an Avian Herpesvirus Reveals Extensive Intrapopulation Diversity in Tandem Repeat Regions, Renner DW. et al. addressed most of the concerns raised by the reviewers in the discussion section. The revision does not have any new data or significant improvement.

Overall, this manuscript provided significant detail on the difference within the tandem repeat regions among different strains of MDV.  However, it did not highlight the key features of tandem repeats that are unique to different strains. Fig. 1D is not informative about the heterogeneity of tandem repeats between attenuated strains and virulent strains. Fig. 1D could be used to summarize the differences in tandem repeats unique for different strains investigated here. Additionally, the discussion section could include a synthesis section on the various tandem repeats. Are there tandem repeats that could be used to predict the virulence of the virus?  What kind of tandem repeats could be seen in attenuated or virulent MDV? What is the driving force for the heterogeneity in the tandem repeats between different strains or within a specific strain?

Reviewer #3: The revised manuscript is improved, and the authors addressed my concerns. There are a few minor issues with the revised manuscript.

Sentence 532-534, “In addition to observing a wide range of copy numbers (2-37 copies) in CVI988, we found that no length was supported by more than 12% of the reads mapping to either locus.” I find this statement difficult to interpret. Is there a better way of stating this?

Sentence 568-570, “Notably, our new model suggests that both of the proline-rich regions found in the MDV genome (i.e., UL36-PRR, Meq- PRR) exhibit complex repetitive patterns that are highly variable across MDV strains.” Please expand on this new model. Perhaps a figure is needed. Proline rich region variations in UL36 and RLORF 7 (meq) have been known for some time.

For meq, (please add the following reference):

Shamblin CE, Greene N, Arumugaswami V, Dienglewicz RL, Parcells MS. Comparative analysis of Marek's disease virus (MDV) glycoprotein-, lytic antigen pp38- and transformation antigen Meq-encoding genes: association of meq mutations with MDVs of high virulence. Vet Microbiol. 2004 Sep 8;102(3-4):147-67. doi: 10.1016/j.vetmic.2004.06.007. PMID: 15327791.

For UL36 (already added)

Spatz SJ, Silva RF. Sequence determination of variable regions within the genomes of gallid herpesvirus-2 pathotypes. Arch Virol. 2007;152(9):1665-78. doi: 10.1007/s00705-007-0992-3. Epub 2007 Jun 8. PMID: 17557133.

Besides these minor issues, the data is well presented in a well written manuscript with beautiful figures.

**Part II – Major Issues: Key Experiments Required for Acceptance**

Reviewer #1: None.

Reviewer #2: No new experiment is requires.

Reviewer #3: none

**Part III – Minor Issues: Editorial and Data Presentation Modifications**

Reviewer #1: None.

Reviewer #2: Fig 1D should be revised to summarize the differences in tandem repeats unique for different strains investigated here.

Reviewer #3: minor issues with their meq and ul36 new model

PLOS authors have the option to publish the peer review history of their article (what does this mean? ). If published, this will include your full peer review and any attached files.

**Do you want your identity to be public for this peer review?** For information about this choice, including consent withdrawal, please see our Privacy Policy .

Reviewer #1: No

Reviewer #2: No

Reviewer #3: No

**Figure resubmission:**
---

## [Editor Report · Decision Letter 1]

6 Aug 2025

Dear Moriah,

We are pleased to inform you that your manuscript 'High-Fidelity Long-Read Sequencing of an Avian Herpesvirus Reveals Extensive Intrapopulation Diversity in Tandem Repeat Regions' has been provisionally accepted for publication in PLOS Pathogens.

Best regards,

Deepak Shukla

Academic Editor

PLOS Pathogens

Robert Kalejta

Section Editor

PLOS Pathogens

Sumita Bhaduri-McIntosh

Editor-in-Chief

PLOS Pathogens

orcid.org/0000-0003-2946-9497

Michael Malim

Editor-in-Chief

PLOS Pathogens

orcid.org/0000-0002-7699-2064
---

## [Editor Report · Acceptance letter]

Dear Dr. Szpara,

We are delighted to inform you that your manuscript, "High-Fidelity Long-Read Sequencing of an Avian Herpesvirus Reveals Extensive Intrapopulation Diversity in Tandem Repeat Regions," has been formally accepted for publication in PLOS Pathogens.

Best regards,

Sumita Bhaduri-McIntosh

Editor-in-Chief

PLOS Pathogens

orcid.org/0000-0003-2946-9497

Michael Malim

Editor-in-Chief

PLOS Pathogens

orcid.org/0000-0002-7699-2064